# Immunological Responses to Seoul Orthohantavirus in Experimentally and Naturally Infected Brown Rats (*Rattus norvegicus*)

**DOI:** 10.3390/v13040665

**Published:** 2021-04-12

**Authors:** Shumpei P. Yasuda, Kenta Shimizu, Takaaki Koma, Nguyen Thuy Hoa, Mai Quynh Le, Zhuoxing Wei, Devinda S. Muthusinghe, Sithumini M. W. Lokupathirage, Futoshi Hasebe, Tetsu Yamashiro, Jiro Arikawa, Kumiko Yoshimatsu

**Affiliations:** 1Department of Microbiology and Immunology, Faculty of Medicine, Hokkaido University, Sapporo 060-8638, Japan; yasuda-sp@igakuken.or.jp (S.P.Y.); kshimizu@med.hokudai.ac.jp (K.S.); j_arika@med.hokudai.ac.jp (J.A.); 2Graduate School of Medicine, Hokkaido University, Sapporo 060-8638, Japan; tkoma@tokushima-u.ac.jp; 3National Institute of Hygiene and Epidemiology, Hanoi 100000, Vietnam; thuyhoa170754@gmail.com (N.T.H.); lom9@hotmail.com (M.Q.L.); 4Graduate School of Infectious Diseases, Hokkaido University, Sapporo 060-0818, Japan; lamtuanglavaron@gmail.com (Z.W.); devindasm@med.hokudai.ac.jp (D.S.M.); sithuminilokupathirage@czc.hokudai.ac.jp (S.M.W.L.); 5Institute of Tropical Medicine, Nagasaki University, Nagasaki 852-8523, Japan; rainbow@nagasaki-u.ac.jp; 6Department of Bacteriology, Graduate School of Medicine, University of the Ryukyus, Okinawa 903-0213, Japan; tyamashi@med.u-ryukyu.ac.jp; 7Institute for Genetic Medicine, Hokkaido University, Kita-ku, Kita-15, Nishi-7, Sapporo 060-0815, Japan

**Keywords:** hemorrhagic fever with renal syndrome, bunyavirus, IgM, CTLs, reservoir

## Abstract

To clarify the mechanism of Seoul orthohantavirus (SEOV) persistence, we compared the humoral and cell-mediated immune responses to SEOV in experimentally and naturally infected brown rats. Rats that were experimentally infected by the intraperitoneal route showed transient immunoglobulin M (IgM) production, followed by an increased anti-SEOV immunoglobulin G (IgG) antibody response and maturation of IgG avidity. The level of SEOV-specific cytotoxic T lymphocytes (CTLs) peaked at 6 days after inoculation and the viral genome disappeared from serum. In contrast, naturally infected brown rats simultaneously had a high rate of SEOV-specific IgM and IgG antibodies (28/43). Most of the IgM-positive rats (24/27) had the SEOV genome in their lungs, suggesting that chronic SEOV infection was established in those rats. In female rats with IgG avidity maturation, the viral load in the lungs was decreased. On the other hand, there was no relationship between IgG avidity and viral load in the lungs in male rats. A CTL response was not detected in naturally infected rats. The difference between immune responses in the experimentally and naturally infected rats is associated with the establishment of chronic infection in natural hosts.

## 1. Introduction

Small mammals such as rodents, shrews, and bats are the natural hosts of viruses of the subfamily *Mammantavirinae*, family *Hantaviridae*, order *Bunyavirales* [1]. Several species of rodent-borne hantaviruses cause two severe zoonotic diseases named hemorrhagic fever with renal syndrome (HFRS) [2], and hantavirus pulmonary syndrome [3]. Humans are infected with these viruses from inhalation of aerosolized excreta of chronically infected reservoir rodents. Hantaviruses are maintained in their specific natural hosts. Both phylogenies of hantaviruses and their host animals were closely correlated, and researchers believe that hantaviruses have coevolved with their host animals [4]. Furthermore, ancient and recent host-switching events might have caused the emergence of different species of hantavirus [5,6].

Seoul orthohantavirus (SEOV) is one of the causative agents of HFRS [7]. The natural host rodent of SEOV is *Rattus norvegicus* (brown rat or Norway rat). Since the *R. norvegicus* life cycle is highly associated with human beings, this species of rats is found worldwide. With the worldwide distribution of host rats, including wild rats, laboratory rats and pet rats, SEOV has also become distributed worldwide and has caused HFRS outbreaks in several countries [8,9,10,11,12,13]. The virus is maintained in rat colonies by horizontal infection from chronically infected rats to susceptible rats [14]. A longitudinal epizootiological study in urban rats showed that infant rats were protected from SEOV transmission by maternal antibodies. After the decline in maternal antibody levels, SEOV is horizontally transmitted to young adult rats from rats with chronic SEOV infection [15]. Naturally infected rats maintain SEOV in their lungs without showing any clinical symptoms [16]. Interestingly, despite the presence of neutralizing antibodies in the blood, SEOV infection is chronically maintained in rats [17].

On the other hand, chronic SEOV infection causing horizontal transmission, as observed in natural hosts, cannot be induced by inoculation of SEOV into conventional laboratory rats [18,19]. The mechanism by which virus infection is maintained in natural host rats remains unknown, due to a lack of research on the immunological status of naturally infected wild rats. The aim of this study was to clarify the mechanism of persistent hantavirus infection in natural host populations. For this purpose, we firstly compared the immune responses to SEOV in experimentally inoculated rats and naturally infected rats.

## 2. Materials and Methods

### 2.1. Virus and Cells

The prototype SEOV strain SR-11 was used in this study [20]. The virus was propagated in Vero E6 cells (obtained from American Type Culture Collection, VERO C1008, ATCC^®^ CRL-1586™), and the culture supernatant was collected. The virus stock was dispensed into vials and stored at −80 °C until use.

### 2.2. Experimental Infection in Laboratory Rats

Six-week-old WKAH/hkm rats (SLC, Hamamatsu, Japan) were inoculated intraperitoneally with SEOV (6 × 10^4^ focus forming units (FFU)/animal). Inoculation dose and route were determined according to the previous studies [21,22]. An outline of the experiments is shown in Appendix A. Two male rats were inoculated with SEOV, then serum specimens were collected from the tail vein at the time of inoculation (day 0) and at 3, 6, 9, 13, 16, 19, 23, 27, 34, 40 and 49 days after inoculation (Appendix A). A total of 34 male rats and 16 female rats were inoculated with SEOV and the spleens, lungs, and sera were collected at different post-inoculation days (Appendix A). Four 8-week-old male Slc:Wistar rats and four 8-week-old male Slc:WistarHannobor/RCC rats were inoculated intraperitoneally with SEOV (6 × 10^4^ FFU/animal). Four rats of each of those two strains were mock-inoculated for controls (Appendix A). Animal experiments were performed after obtaining permission from the Institutional Animal Care and Use Committee of Hokkaido University (08-0374). Experiments involving virus infections were performed in a biosafety level 3 (BSL-3) facility.

### 2.3. Collection and Analyses of Brown Rats of SEOV

A total of 199 brown rats (*Rattus norvegicus*) were captured in the period from 2009 to 2012 in the Hanoi City and Haiphong Port areas in Vietnam, where SEOV-positive brown rats were previously captured [23]. Rat trapping was carried out for pest-control purposes by the staff of Haiphong International Port Quarantine Station at several warehouses in Haiphong Port. Subsequently, we purchased those rats. Rat trapping was also carried out in the backyard of the National Institute of Health and Epidemiology located in Hanoi City. Rats were captured by using cage-type live traps with roasted coconut or apple as bait. Blood samples were collected by cardiac puncture under anesthesia and sera were separated. After collecting blood samples, rats were euthanized by cervical dislocation. Sex, body weight and lengths of the head and body, tail, ear, skull and hind foot pad of the rats were recorded. Spleens were collected and soaked in Dulbecco’s modified Eagle medium (Thermo Fisher Scientific, Waltham, MA, USA) supplemented with 10% fetal bovine serum (Thermo Fisher Scientific) for splenocyte isolation. Lung tissues and feces in the colon were preserved on ice for viral genome detection, and eyeballs were collected and soaked in 10% formalin in phosphate-buffered saline (PBS) for age estimation [24].

### 2.4. Antibody Detection in Rat Sera

Anti-hantavirus immunoglobulin G (IgG) and immunoglobulin M (IgM) antibodies in rat sera were detected by ELISA as described previously [25]. Because wild rat sera showed a high background reaction in ELISA for IgM detection, a Western blot assay was performed to detect IgM as described previously for detecting rat serum IgM antibodies with low cross-reactivity of IgG antibodies [26,27]. Briefly, the recombinant N protein of SEOV, expressed in High Five cells by a baculovirus vector, was used as the Western blot antigen [25]. Horseradish peroxidase-labeled goat anti-rat IgM antibody (Kierkegaard & Perry Laboratories, Inc., Gaithersburg, MD, USA) was used as a secondary antibody. IgG avidity was measured by comparison of ELISA OD values obtained with and without washing with 6 M urea solution, as previously described [21]. 

### 2.5. Quantification of Viral RNA

Total RNA was extracted from sera, lungs, and feces by Isogen-LS and Isogen (Nippon Gene, Tokyo, Japan) following the manufacturer’s instructions. It was reversely transcribed into cDNA by using random primers and SuperScript^®^ II Reverse Transcriptase (Thermo Fisher Scientific) according to the manufacturer’s instructions. The cDNAs were subjected to real-time PCR analysis, using primers SEOS_F (5′-tatggttgcctggggaaag-3′), SEOS_R (5′-gctctggatccatgtcatca-3′), and Universal Probe Library #86 (gcagtgga, Roche, Basel, Switzerland), by using Light Cycler 480 Probes Master and Light Cycler 480 Instrument II (Roche) according to the manufacturer’s instructions (Appendix A). 

### 2.6. Detection of SEOV-Specific Cytotoxic T Lymphocytes (CTLs) in Inbred WKAH/hkm Rats

In total, 225 15-mer peptides with 10 aa overlaps, encompassing the entire amino acid sequence of the envelope glycoprotein of the SEOV strain SR-11 and 84 15-mer peptides for N protein, were purchased (Mimotopes Pty Ltd., Mulgrave, Australia). Twenty peptide pools, each containing 16 peptides, were dissolved in dimethyl sulfoxide. Splenocytes were obtained by using lympholyte^®^-Rat according to the manufacturer’s instructions (Cedarlane Laboratories Ltd., Burlington, ON, Canada). A mixture of 5 × 10^5^ spleen cells from a single laboratory rat or 10^5^ spleen cells from a single wild brown rat, 3.125 µg/mL peptide, and 20 ng/mL of rat interleukin-2 was cultured in the chamber of Rat interferon-gamma ELISPOT^®^ according to the manufacturer’s instructions (Cellular Technology Limited, Cleveland, OH, USA). After screening by an ELISPOT assay of 20 peptide pools, epitope peptides were determined by another ELISPOT assay using a single peptide solution. The peptides derived from envelope glycoprotein regions #70 (QSVCDNNALPLIWRG) and #149 (QYPWHTAKCHFEKDY), and from N protein region #286 (SPSSIWVFAGAPDRC) were identified as SEOV-specific CTL epitopes of WKAH/hkm (MHC RT1k) rats. By using #149 peptide, which induced the largest number of ELISPOT foci, and #1 peptide (MWSLLLLAALYGQGF), which is the 1–15 region of Gn, as a negative control, fluctuation of CTLs after inoculation was examined in WKAH/hkm rats.

### 2.7. Detection of SEOV-Specific CTLs in Experimentally Inoculated Outbred Rats and Naturally Infected Wild Brown Rats 

Spleen cells (5 × 10^5^ cells) from laboratory outbred Wistar and Wistar-Hannover rats and spleen cells (10^5^ cells) from wild rats were used for ELISPOT assays, as described above. Colored areas were quantified using Image J software. The colored area developed by each of the 20 pools was measured, and then the average value was calculated. Next, the activation index was calculated by comparing the average value to the colored area developed using dimethyl sulfoxide, which was the solvent without the peptide. Six matured brown rats, including three males and three females (>300 g in weight, mean age of 17 months), were used in this study. They had both anti-SEOV IgG and IgM antibodies and viral genomes in their lungs. Therefore, these rats were considered to be rats chronically infected with SEOV. Sixteen brown rats that were negative for the antibody and viral genome were used as a negative group (>286 g in weight, mean age of 12 months). The mean age of total rats used in this study was 12.6 months. The mean age of viral genome-positive rats was 19.8 months, and that of viral genome-negative rats was 10.4 months. Although there was an age bias between the two groups, it was a bias derived from the original population.

### 2.8. Statistical Analysis

The chi-square test was used to examine the relationships of IgM antibody positivity and viral load in the lungs or serum, and the activation index of CTL activity in SEOV-infected rats. The correlation test was used to examine the relationship between IgG avidity and viral load in the lungs [28].

## 3. Results

### 3.1. Immune Response of Experimentally Infected Rats

In the experimental infection, the rat antibody response showed a transient increase in IgM, a subsequent increase in IgG, and then an increase in IgG avidity (Figure 1A). Novel and dominant CTL epitopes of inbred rats of the WKAH/hkm strain were identified by the ELISPOT system. As a result of analysis using the #149 peptide (QYPWHTAKCHFEKDY), the peak of CTLs was 6 days after inoculation (Figure 1B). In addition, it was confirmed that the virus was almost eliminated from the blood after the transient rise in CTLs, which are considered to be effector T cells (Figure 1C). The virus tended to remain in the lungs at a low level. A significant sex difference in viral load was not found. IgM and IgG responses in rats used for the CTL assay and viral RNA detection are shown in Appendix A. 

### 3.2. Immune Response of Naturally Infected Rats

#### 3.2.1. Antibody Production 

As shown in Table 1, 43 of the 199 rats were IgG antibody-positive for SEOV. There was only one rat that carried only IgM, which was considered to be at the very early phase of infection. Of the 43 IgG-positive rats, 28 rats (65%) were IgM-positive, suggesting the presence of long-lasting IgM antibodies in natural host rats. A sex difference in the positivity of IgG and IgM antibodies was not found (Appendix A). Two seropositive juvenile female rats, 0.93 and 1.2 months old, respectively, were eliminated from further analyses because they seemed to be carrying antibodies received from their mothers (data not shown). On the other hand, sex differences were found in IgG avidity and IgM positivity rates (Table 2). We tentatively categorized samples into acute and chronic samples according to the IgG avidity percentage. Significant sex differences (*p* < 0.05) were found only in the acute phase. In male rats, the IgM positivity rate in the chronic phase was higher than that in the acute phase (*p* < 0.05). In female rats, as maturation of IgG avidity occurred, the IgM positivity rate decreased. However, a significant difference was not found (*p* = 0.24).

#### 3.2.2. Quantification of the Viral Genome and Comparison with the Immunological Status

A comparison of the viral genome abundance in the lungs showed that the quantity of the viral genome tended to decrease as IgG avidity increased in females (*r* = −0.696, *p* = 0.001) (Figure 2). On the other hand, in males, there was no association between increments in IgG avidity and viral load in the lungs (*r* = 0.088, *p* = 0.721). The virus was maintained in the lungs of the rats. Viral loads in the lungs of wild rats were higher than those in experimentally inoculated rats (see Figure 1C). Next, we examined IgM antibody positivity and viral genome loads in the lungs (Table 3). However, a significant relationship was not found by an independence test (*p* = 0.125). IgM antibody levels and viral genome loads in sera were also not significantly related (Appendix A). Although no significant difference was detected, rats carrying IgM antibodies tended to have persistently high viral loads in their lungs and serum. We also examined the relationship of viral genome loads in sera and feces. As shown in Figure 3, in rats with viral genome loads exceeding 10^5^ copies/mg in their lungs, the virus was detected in feces and blood. These results indicated that high viral loads in the lungs might cause virus shedding in feces and serum.

#### 3.2.3. Cellular Immune Response

Next, we tried to detect cellular immune responses in naturally infected rats. An ELISPOT assay for inbred rats has already been established as mentioned above (Figure 1B). First, we tried to apply this assay to outbred rats. As shown in Figure 4, the activation index was significantly increased in virus-inoculated outbred rats. In wild rats, varied activation indexes were observed even in SEOV-negative rats. Six naturally infected wild rats were selected and examined by the same method. As shown in the right panel of Figure 4, no difference was observed between chronically SEOV-infected rats and non-infected rats. It can be considered that significant cell-mediated immunity is not induced or maintained in wild rats.

## 4. Discussion

It was reported that rodent hosts of hantaviruses are apparently healthy and possess neutralizing antibodies, but high titers of hantaviruses were found in the lungs at the same time [14]. We previously studied hantavirus persistence using laboratory mice and a prototype orthohantavirus, Hantaan virus (strain 76–118). As a result, we succeeded in producing chronically infected mice by using two methods: experimental infection of newborn mice with a sublethal dose [29] and splenocyte transfer to infected SCID mice [30]. Both models showed suppression of CTLs, indicating prevention of self-attack by CTLs in virus-infected lung tissue. From these results, we hypothesized that the cell-mediated immune response is suppressed in the natural host without suppression of humoral immunity. In this study, we attempted to clarify the state of viral infection by analyzing humoral immunity and cell-mediated immunity in rats naturally infected with SEOV.

In the experimentally inoculated rats, the immune response corresponded to the typical pattern in transient and acute viral infection. The transient appearance of IgM antibodies was followed by the appearance of IgG antibodies and the transient appearance of CTLs. The ELISPOT system also confirmed that the virus was eliminated from the blood 6 days after the transient appearance of SEOV-specific CTLs, which were considered to be effector T cells and also to be involved in the rapid elimination of the virus from serum. However, a low level of the viral genome was continuously detected in the lungs, suggesting that the specific CTLs might not be effective in eliminating the virus from the lungs. 

Hanoi Province and suburban ports in Vietnam were selected as sites of natural host colonies of brown rats for the following reasons. Firstly, we had been conducting continuous SEOV epidemiological studies in those areas [23]. Secondly, our peptide library, based on the SEOV strain SR-11, could be used for the analysis of the cellular immune response to Vietnamese SEOV, due to the high sequence similarities [23,31]. Amino acid sequence similarities of Vietnamese SEOV to SR-11 were estimated to be 98.4% (N) and 99.1% (GP). In the amino acid comparison of N protein, most of the amino acid differences were congregating toward amino acid numbers 68–74, and only one amino acid difference was found in the remaining region. An RT1k-restricted T-cell epitope on GP, QYPWHTAKCHFEKDY, was identified in this study. It was found to be conserved among SEOV strains derived from Hanoi, Haiphong and Saigon ports. For real-time PCR analysis, primer and probe sequences were also conserved among these strains. Thirdly, the availability of a BSL3 facility in the National Institute of Hygiene and Epidemiology in Hanoi was important for the sample handling, rat spleen cell separation, and ELISPOT assay. 

IgG avidity is generally thought to be matured as time passes after hantavirus infection in patients [32]. Previous studies suggested that males might be more important than females for virus spreading in a rat population. The immune response to SEOV in laboratory rats differs depending on sex, due to the differences in behavior and innate immune responses [33,34,35]. According to a study in wild rats, wounding among males was reported to be more important than inhalation for horizontal transmission [36]. However, the details of immunological responses in natural host rats remain unclear. The results of this study suggested that atypical immune responses may occur in male rats in nature. 

The viral load in the lungs of naturally infected rats was higher than that in experimentally infected rats. SEOV RNA was detected in feces and blood when the virus genome exceeded 10^5^ copies/mg in the lungs. Feces might act as a source of horizontal SEOV infection among rats. In contrast, experimentally inoculated rats with a low copy number of SEOV in their lungs probably rarely exceed this criterion. The burden on the lungs and immune system due to coinfections of mycoplasma and other microbes or cancer-bearing cells may be involved in the high viral load of SEOV and efficient virus spread by coughs in laboratory rats under inappropriate conditions [37,38]. 

There are some inadequacies in this study. Initially, there is the issue of virus strains and transmission routes in natural hosts. Differences in infection patterns between a Vero cell-adapted virus and a host-passaged virus have been shown with the combination of Puumala orthohantavirus and bank voles [39]. However, no brown rat-passaged SEOV strain has been reported. The combination of Vero cell-adapted virus and immunocompetent laboratory rats provides low levels of viral persistence that are inadequate for horizontal transmission. This indicates that either the cell-adapted virus or the experimental rat might be responsible for the inability to reproduce chronic infections observed in nature. To establish a rat-passaged SEOV, we tried to find antibody-negative wild rats with the virus in their lungs to rule out the effects of neutralizing antibodies for passage. Two virus-positive rats were found among 96 antibody-negative rats, and their lung homogenates were intranasally inoculated into laboratory rats. However, 49 days after inoculation, no seroconversion was detected and the virus passage by rats was unsuccessful. Therefore, in this study, we performed experimental inoculation using SR-11. Another approach needs to be implemented to prepare an appropriate virus strain. 

The average age of wild rodents used in this study was 12.6 months. On the other hand, the experimental rats used in this study were younger. In this study, 8-week-old experimental rats were used because they were well-nourished and considered to be adults. However, it is necessary to consider the age of rats for the establishment of chronic SEOV infection. 

In addition, in order to reproduce an experimental infection close to that in natural hosts, it is necessary to consider the inoculation route. The results of a study using the combination of Sin Nombre virus (SNV) and deer mice have been reported [40]. Several routes of infection were used in that study, and it was found that intraperitoneal inoculation resulted in more stable infection. In an experimental infection of SR-11 conducted in the 1990s, intraperitoneal inoculation showed more stable seroconversion than did intramuscular or intranasal inoculation (personal communication). Therefore, many experiments using SR-11 were conducted with the intraperitoneal route of infection [19,22,41]. In this study, we also intraperitoneally inoculated rats with SEOV, but future studies should compare immune responses induced by various inoculation routes. Subcutaneous inoculation, assuming a bite and intranasal inoculation, assuming aerosol infection, should be examined. In the study of SNV, the efficiency of transmission to cage mate increases with stress, and it can therefore be considered that the host factor is involved in the efficiency of horizontal transmission. It is necessary to determine factors that are involved in the establishment of horizontal transmission in the combination of SEOV and brown rats. On the other hand, in the infection model of SNV-infected deer mice, almost no difference in seroconversion and viral loads was found between male and female deer mice after SNV inoculation, being different from the results obtained for SEOV and rats. The mechanism of the coexistence of orthohantavirus with rodents may vary.

In this study, we tried to detect CTLs in natural hosts. As a result, no difference was observed between the virus-negative group and the virus-positive group, indicating that a detectable CTL level is not maintained in naturally infected rats. Continuous antigenic stimulation due to chronic infection of SEOV may cause continued IgM production and suppression of CTL induction. However, determination of the activation index using pooled peptides was not a sensitive method because two of the 8 outbred laboratory rats showed a lower activation index. The methodology for detecting CTLs from wild rats should be improved. If CTLs are continuously induced in rats during chronic infection, they would affect the health status by attacking lung tissue. Both the virus and host might be able to survive under the condition of depleted SEOV-specific CTLs. Conversely, suppression of CTL induction can also cause chronic infection. Regulatory T-cells might play an important role in the control of CTLs, even in nature [42]. We tested the CTL activities of six rats with viral RNA in their lungs. Among them, only rat #188 was considered to contribute to virus spreading, because that rat possessed viral RNA in its blood and feces and it showed the lowest CTL activation index of 0.51. Accumulation of data from virus-shedding rats like rat #188 is needed to understand the chronic infection of SEOV in reservoir rats.

The phenomenon in which a hantavirus is nonpathogenic to its host but pathogenic to humans is common to various zoonoses. The immunomodulatory mechanism that causes selective immunosuppression in rodents might be directly linked to the pathogenicity expression mechanism in humans. Further studies on natural hosts of hantaviruses are needed to understand its pathogenicity and for the control of hantavirus infection.

## Figures and Tables

**Figure 1 viruses-13-00665-f001:**
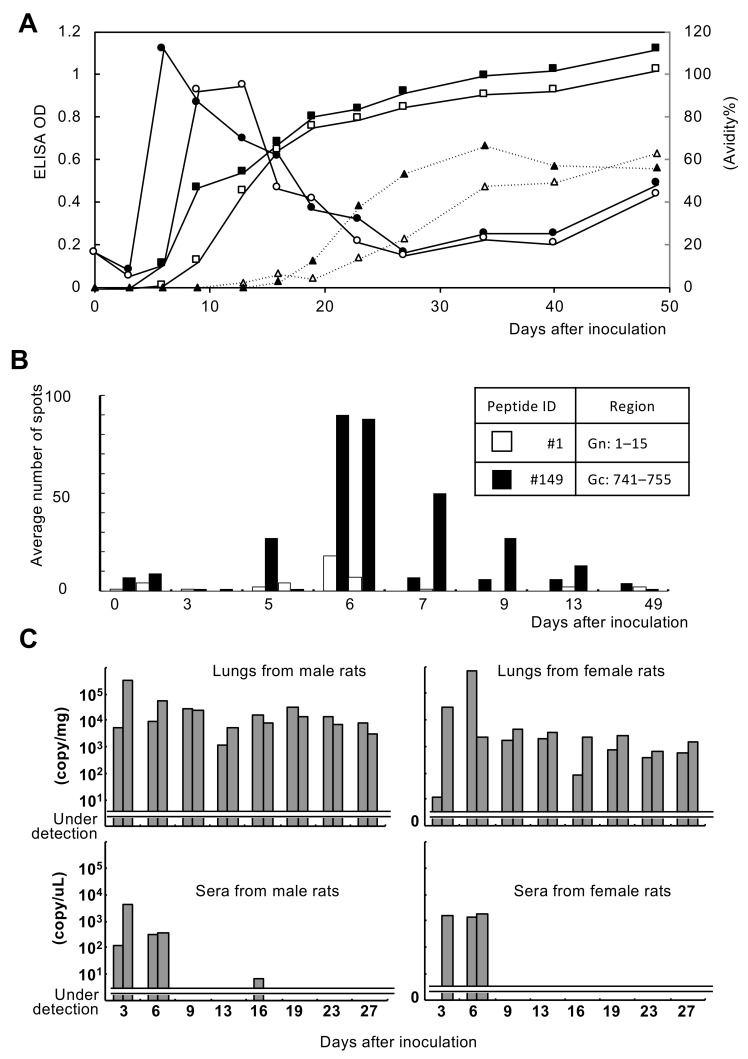
Immunological responses and virus replication in Seoul orthohantavirus (SEOV)-inoculated WKAH/hkm rats. (**A**) Fluctuations of IgM and IgG antibodies and IgG avidity index in rat sera. Circles, ELISA OD for IgM antibody detection; squares, ELISA OD for IgG antibody detection; triangles, IgG avidity %. Open markers are results from rat #1 and closed markers are results from rat #2. (**B**) Results of the ELISPOT assay. Two rats were examined by the ELISPOT assay by using control peptide #1 and epitope peptide #149. (**C**) Viral replication in lungs and sera was determined by a real-time PCR assay. Two rats were examined at each time point.

**Figure 2 viruses-13-00665-f002:**
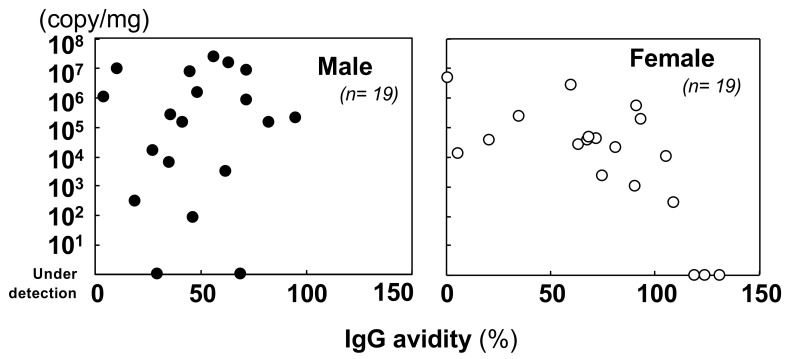
Comparison of IgG avidity and viral load in lung tissues of male and female wild rats.

**Figure 3 viruses-13-00665-f003:**
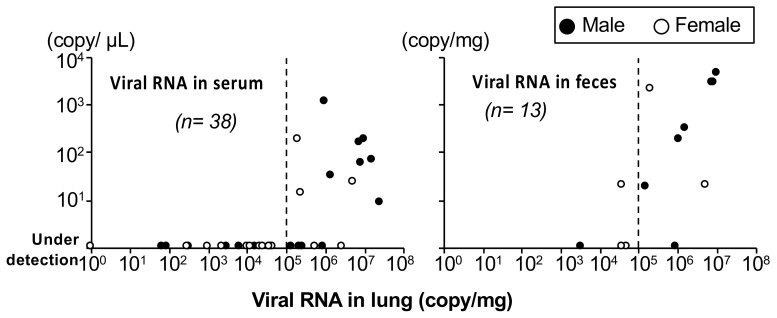
Relationship between virus genome copies in lung tissues and those in serum or feces of wild rats. Rats with viral genome loads exceeding 10^5^ copies/mg in their lungs tended to have the genome in serum and feces. Male rats had more viral genomes in serum and feces than did female rats.

**Figure 4 viruses-13-00665-f004:**
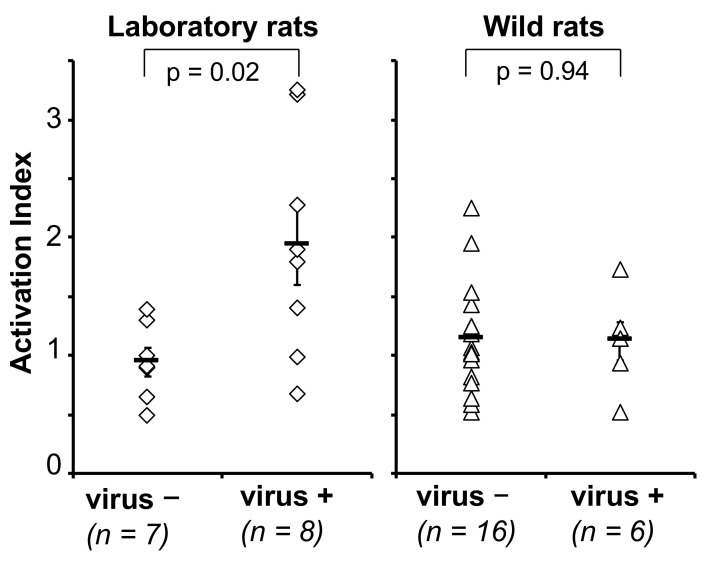
Detection of cytotoxic T lymphocyte activities in laboratory rats experimentally inoculated with SEOV, and wild rats naturally infected with SEOV.

**Table 1 viruses-13-00665-t001:** Numbers of anti-SEOV IgM and IgG antibody-positive wild rats.

All Specimens	IgG +	IgG −	Total
IgM +	28	1	29
IgM −	15	155	170
Total	43	156	199

**Table 2 viruses-13-00665-t002:** IgG avidity and IgM positivity rate in IgG antibody-positive wild rats.

Phase	IgG Avidity (%)	IgM Positive Rate (%)	
		Male	Female *
Acute	≤30	1/5 (20%)	3/3 (100%)
Chronic	<30–≤60	5/7 (71%)	2/2 (100%)
	<60–≤90	6/7 (86%)	6/7 (86%)
	>90	1/1 (100%)	4/9 (44%)

* Two juveniles were excluded from the analysis.

**Table 3 viruses-13-00665-t003:** SEOV genome detection in lungs of IgG-positive wild rats.

Viral Genome in Lung	IgM Antibody in Sera *	
	+	−
+	24	9
−	3	4

* Because one lung sample was lost, 40 samples were analyzed.

## Data Availability

The data presented in this study are available on request from the corresponding author.

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
