# Peer review of "Immunological Responses to Seoul Orthohantavirus in Experimentally and Naturally Infected Brown Rats (Rattus norvegicus)"

_viruses, 2021, doi:10.3390/v13040665_

Round 1
Reviewer 1 Report
Viruses-1096783
Immunological response to Seoul orthohantavirus in experimentally inoculated rats and naturally infected rats (Rattus norvegicus)
Overall Evaluation:
Dr. Kumiko Yoshimatsu and her colleagues have compared the humoral and cell-mediated immune responses of brown rats (Rattus norvegicus) to experimental and natural infection with Seoul orthohantavirus (family Hantaviridae). Experiments are well described and analyses are appropriate. However, concerns about the design of this preliminary study diminishes overall enthusiasm.
Major Comments:
Materials and Methods, page 2, line 69: The SR-11 strain was isolated many decades ago from a laboratory rat (Rattus norvegicus) implicated in a laboratory outbreak of HFRS in Sapporo, Japan, and represents a well-known laboratory-adapted strain of SEOV that has been passaged multiple times in Vero E6 cells. As such, using this SEOV strain in experimental infection might not correspond to what actually happens in naturally acquired SEOV infection in brown rats. The authors do not discuss if this might account for the vast differences in immunological responses to SEOV infection. And it is unclear if the investigators contemplated using lung homogenates from naturally infected rats as the inoculum in experimental infection studies.
Materials and Methods, page 2, lines 73 and 78: Six-week-old WKAH/hkm rats, 8-week-old male Slc:Wistar rats and 8-week-old male Slc:WistrarHanorbor/RCC rats were used in experimental infection studies. But the authors do not explain the basis for the age of rats selected. Nor do they provide data on the age at which brown rats typically acquire SEOV infection in nature. Also, what was the basis for the selection of rat strains? And did the authors consider brown rats of different ages for their experimental infection studies?
Materials and Methods, page 2, lines 73: The authors refer to previous studies suggesting that SEOV infection in male rats might be acquired through wounding. However, the intraperitoneal route of inoculation was used. The authors do not discuss whether the intraperitoneal route of infection accounted for the observed differences in immunological responses. That is, would alternative routes of experimental infection (such as intramuscular or intranasal) yield immunological responses more akin to natural infection?
Materials and Methods, page 2, lines 74: Each rat was inoculated intraperitoneally with 6 x 104 FFU of SEOV strain SR-11. The authors do not explain the basis for selecting this dose. It is not altogether clear if this represents the ID50 in the rat strains used.
Materials and Methods, page 3, lines 115–116: Peptides for CTL studies were all based on SEOV strain SR-11. Although the authors claim that SR-11 exhibits “high sequence similarity” with SEOV strains from Vietnam, it is not clear if SEOV strains from brown rats captured in Hanoi and Haiphong have been fully sequenced.
Discussion, page 8, line 222: The authors should discuss the limitations of their study design.
Minor Comments:
Title, page 1, lines 2–4: The authors should consider revising their title to read: Immunological responses to Seoul orthohantavirus in experimentally and naturally infected brown rats (Rattus norvegicus)
Abstract, page 1, lines 21–23: The authors should consider revising their first sentence to read: To clarify the mechanism of Seoul orthohantavirus (SEOV) persistence, we compared the humoral and cell-mediated immune responses to SEOV in experimentally and naturally infected brown rats.
Introduction, page 1, line 42: Reservoir hosts are not “contaminated”. So “contaminated rodents” should be revised to “chronically infected reservoir rodents”.
Introduction, pages 1–2, lines 44–46: The authors may wish to revise their statement about hantavirus evolution and cite the paper by Ramsden and colleagues, regarding the lack of codivergence (Mol Biol Evol. 2009;26:143-153).
Materials and Methods, page 3, line 95: The eyes of wild rats were preserved to estimate the age of the rodent, but immunological data were not presented according to the age of the rats.
Discussion, page 8, lines 223–224: The first sentence in the Discussion section is awkward. The authors state: Natural host animals of hantaviruses maintain a large number of viruses in their lungs, while possessing neutralizing antibodies at the same time, and are apparently healthy [13].” Hantavirus reservoir host are not known to “maintain a large number of viruses in their lungs”. Instead, high titers of hantavirus are typically found in the lungs of reservoir rodent hosts.
Figure 1C, page 5: “Sera from mail rats” should be “Sera from male rats”.
Table 2, page 6: “90<” is incorrect; it should be “>90”.
References, pages 10–12: The formatting of the references does not conform to the journal instructions. Specifically, some titles are in upper case.
Reviewer 2 Report
Line 42. Infected rodents.
Line 168 and 180. Round to whole numbers because 1/10 %s where Ns are only 43 or much less are insignificant and implies much larger Ns. Rounding also simplifies the tables and does not alter the interpretation of the results.
267-268. Not a complete sentence.
SEOV is an important pathogen and occurrence of human infection is doubtless significantly underreported. Although a pilot study with relatively small numbers of rats, these results are interesting. Persistence of IgM and infectious virus indicates chronic infection. Do the authors know in which cells in the lungs the virus is present and persists?
Author Response
Answers to reviewer #2
Article number viruses-1096783.
Immunological response to Seoul orthohantavirus in experimentally inoculated rats and naturally infected rats (Rattus norvegicus)
Line 168 and 180. Round to whole numbers because 1/10 %s where Ns are only 43 or much less are insignificant and implies much larger Ns. Rounding also simplifies the tables and does not alter the interpretation of the results.
According to your suggestion, we fixed.
267-268. Not a complete sentence.
I modified this part.
SEOV is an important pathogen and occurrence of human infection is doubtless significantly underreported. Although a pilot study with relatively small numbers of rats, these results are interesting. Persistence of IgM and infectious virus indicates chronic infection. Do the authors know in which cells in the lungs the virus is present and persists?
Recent study for SEOV pathology in natural host rats is very fine {Maas, 2019 #22}. I guess, SEOV infects various cells in lung such as alveolar cells, pericytes, and endothelial cells without emerging CPE. However, SEOV induce low level inflammation and are eventually taken up by alveolar macrophages and excreted during exhalation due to coughing. Hantaviruses can survive in macrophage, so that it can be source of infection. In the 1980s, when SEOV outbreaks occurred at animal facilities in Japan, most of facilities were not microbiologically controlled, mycoplasma infections were widespread, and rat were coughing. Cough is probably associated with horizontal transmission of SEOV among domestic rats. Therefore, we modified discussion section in line 299.
Reviewer 3 Report
The authors present a novel study to investigate the parallels between experimentally-infected and naturally-infected reservoir hosts in their immune responses and course of infection with the orthohantavirus Seoul virus. Answering this question will provide needed context for experimental studies of persistent infections in reservoirs as natural hosts of zoonotic RNA viruses. This is an important question to be answered to provide clues for pathogenic outcomes in humans. This study makes progress in understanding this phenomenon but the direct comparison between experimentally infected rats and wild-caught rats requires more information to be meaningful.
Major questions:
- The virus used for experimental inoculation is SR-11, a lab adapted strain that presumably has been passaged many times in vero cells and may be cell culture adapted. It is worth considering that mutations introduced to this virus due to cell culture adaptation could explain the observed differences in host responses.
- Viral copies were detected in lungs out to 27 days in both male and female rats but not in sera. However, the authors conclude that these animals are not persistently infected. This conclusion appears at odds with the data.
- Figure 2 lacks statistical analysis to support the conclusion that viral load decreased as IgG avidity increased in females.
- Figure 4 argues that wild rats do not mount a cytotoxic T cell response, but that experimentally inoculated rats do. Because the authors use IgG avidity as a maker for duration of infection, are the experimental animals matched for IgG avidity compared to wild? Unclear if this lack of cytotoxic T cell response is a natural course of a longer infection or a true difference between wild and lab rats.
- The argument in the discussion about the absence of viremia in the serum of laboratory mice following the rise of CTLs is not terribly convincing since most of the wild mice also had no serum viremia but also no CTL response.
- Statement 267-268 is not a complete statement.
- It’s not clear how many of the captured wild rats were positive for viral RNA. Or was just IgG prevalence investigated?
- Figure 1A is only portraying data from two rats. Difficult to understand how the authors can make conclusions on immunological trends in experimental infections with just two individuals.
- 30% avidity only seen (in the two rats analyzed) at day ~25 post infection. This is getting closer to the “persistent/chronic” infection of the wild rats. If the authors just compared responses at those time points to those of the captured wild rats, what would be the conclusion? Is it possible to stratify the comparisons by “acute” and “persistent”?
- The authors don't mention any parallels to many similar studies in reservoirs of Sin Nombre virus (a new world hantavirus) performed by others. This could be useful for context to the reader. Are there similarities in reservoir responses?
Round 2
Reviewer 1 Report
The authors have responded to the comments and concerns. There are remaining grammatical errors but these presumably can be dealt with at the type-setting stage.
